# Radiological Hazards and Natural Radionuclide Distribution in Granitic Rocks of Homrit Waggat Area, Central Eastern Desert, Egypt

**DOI:** 10.3390/ma15124069

**Published:** 2022-06-08

**Authors:** El Saeed R. Lasheen, Mokhles K. Azer, Antoaneta Ene, Wael Abdelwahab, Hesham M. H. Zakaly, Hamdy A. Awad, Nilly A. Kawady

**Affiliations:** 1Geology Department, Faculty of Science, Al-Azhar University, Cairo P.O. Box 11884, Egypt; elsaeedlasheen@azhar.edu.eg; 2Geological Sciences Department, National Research Centre, 33 El Bohooth St. (Former El Tahrir St.), Dokki, Giza P.O. Box 12622, Egypt; mokhles72@yahoo.com (M.K.A.); dr.wael.nrc@gmail.com (W.A.); 3INPOLDE Research Center, Department of Chemistry, Physics and Environment, Faculty of Sciences and Environment, Dunarea de Jos University of Galati, 47 Domneasca Street, 800008 Galati, Romania; 4Physics Department, Faculty of Science, Al-Azhar University, Assiut Branch, Assiut 71524, Egypt; 5Institute of Physics and Technology, Ural Federal University, Yekaterinburg 620002, Russia; 6Geology Department, Faculty of Science, Al-Azhar University, Assiut Branch, Assiut 71524, Egypt; 7Geochemical Exploration Department, Nuclear Materials Authority, El Maadi, Cairo P.O. Box 530, Egypt; nillykawady@yahoo.com

**Keywords:** natural radioactivity, radiological health assessment, granitic rocks

## Abstract

Natural radioactivity, radiological hazard, and petrological studies of Homrit Waggat granitic rocks, Central Eastern Desert, Egypt were performed in order to assess their suitability as ornamental stone. On the basis of mineralogical and geochemical compositions, Homrit Waggat granitic rocks can be subdivided into two subclasses. The first class comprises granodiorite and tonalite (I-type) and is ascribed to volcanic arc, whereas the second one includes alkali-feldspar granite, syenogranite, and albitized granite with high-K calc alkaline character, which is related to post-orogenic granites. ^238^U, ^226^Ra, ^232^Th, and ^40^K activities of natural radionuclides occurring in the examined rocks were measured radiometrically using sodium iodide detector. Furthermore, assessment of the hazard indices—such as: annual effective dose (AED) with mean values (0.11, 0.09, 0.07, 0.05, and 0.03, standard value = 0.07); gamma radiation index (Iγ) with mean values (0.6, 0.5, 0.4, 0.3, and 0.14, standard value = 0.5); internal (Hin) with mean values (0.6, 0.5, 0.4, 0.3, and 0.2, standard value = 1.0); external (Hex) index (0.5, 0.4, 0.3, 0.24, and 0.12, standard value = 1.0); absorbed gamma dose rate (D) with mean values (86.4, 75.9, 53.5, 43.6, and 20.8, standard value = 57); and radium equivalent activity (Raeq) with mean values (180, 154, 106.6, 90.1, and 42.7, standard value = 370)—were evaluated with the knowledge of the natural radionuclides. The result of these indices falls within the acceptable worldwide limits. Therefore, we suggest that these rocks are safe to be used in industrial applications.

## 1. Introduction

Granitic rocks represent the most abundant plutonic rocks in the continental crust [1]. Their distribution in Arabian-Nubian Shield (ANS) is heterogeneous, since they are abundant in the northern ANS [2,3,4,5,6]. Granitic rocks cover about sixty percent of the Egyptian crystalline rocks in the Eastern Desert, which decreases to thirty percent in the south (Kenya and Ethiopia) of ANS [7,8]. Field, petrological, geochemical, and isotopic studies discriminate the Egyptian granitic rocks into older (850–610 Ma) and younger granites (610–550 Ma) [2,8]. The former (older) include calc-alkaline tonalite-granodiorite association (gray) of syn-orogenic rocks with I-type signature. In contrast, the younger assemblages (610–550 Ma) include pink, calc-alkaline to alkaline with late- (I-type) to post-orogenic (A-type) associations [8,9].

Accordingly, several studies reveal different petrogenesis of Egyptian granitic rocks, varying from syn-late-orogenic to post-orogenic as well as anorogenic granites. Among different granitic rocks, the post-orogenic and anorogenic granites have conspicuous rare metals enrichment that are commonly modified by later hydrothermal processes such as albitization, silicification, and fluoritization [9,10,11,12,13]. This rare metal mineralization includes Zr, Ta, Nb, and W as well as others, such as high contents of rare earth element (REE), U, and Th mineralization, attracting the attention of many geologists and authors to shed light on theses rocks, due to their importance in industrial applications [14].

On the other hand, natural radionuclides, including ^238^U, ^226^Ra, ^232^Th, and ^40^K are widely distributed in most felsic, potassic suites of igneous rocks (granitic rocks) [2,15]. These rocks are most adequate for various natural radionuclide occurrences [2]. There are strong relationships between radionuclides and volatile components such as chlorine and fluorine, where these radionuclides tend to concentrate in the most felsic rocks [2,16]. Cosmic radiation is considered the main source of external radiation and includes approximately 90% of high-energy protons. In addition, external radiation comes from the soil, rock materials, and building materials represented in natural chain decay of uranium series, thorium series, and ^40^K. The assessment of these natural radioactive elements permits estimation of radiation hazard indices and risks to human beings. Therefore, the level of radioactivity in any area is considered as an indicator for the hazard region that affects human health [2,15,16].

The current study aims to integrate field, petrographical, and geochemical studies of Homrit Waggat granites to clarify their different mineralogical composition and tectonic setting. In addition, for the first time, this study discusses the natural radioactivity as well as radiological health risk of different Homrit Waggat magmatic rocks in order to assess their suitability as ornamental stone.

## 2. Geologic Setting

Younger granitic rocks are abundant in Central Eastern Desert of Egypt relative to the southern part [2,10,17]. Multi magmatic phases of Homrit Waggat area lie ~50 km NW of Mersa Alam town (Figure 1a) that have been discussed by many authors, such as Azer et al. [10]. Volcano-sedimentary rocks and metagabbro-diorite complex of island-arc suite represent the oldest rocks in the studied area. The metagabbro-diorite complex exposes in the northern sector of the mapped area (Figure 1b) and intrudes the volcani-sedimentary rocks with sharp contacts.

Homrit Waggat granitic rocks can be classified into older and younger rocks based on geochronological sequence [10]. The former is oval, gray in color, fine-grained, and represented by small exfoliated and separated hills with low relief. They are intruded by dikes of different sizes and shapes. They include small xenoliths of gneiss occurring in several masses. In contrast, the younger granitic rocks are widely distributed, constituting an area of ~50 km^2^ and more than 1 km above sea level. They are exposed in the central part as two isolated wings forming an elliptical ring. They are medium-to-coarse-grained and pink in color; they intrude older granitic rocks and are invaded by pegmatite and quartz veins.

## 3. Material and Methods

Twenty-five samples were collected from the examined area. Fifteen samples were selected for petrographic investigations through thin sections that have been studied using a polarizing microscope. Five representative samples have been selected for bulk-rock geochemical analysis by XRF (X-ray fluorescence) at the National Research Center (NRC) in fused bead of the sample with a ratio of 1 g (sample)/10 g (lithium metaborate 34%/lithium tetraborate 66%) at 1150 °C in an electroconductive furnace. Detection of element concentrations was carried out with a sequential WD-XRF spectrometer and the ASTM D-7348 standard test method for loss on ignition (LOI).

^238^U, ^232^Th, ^226^Ra, and ^40^K were examined for variable rocks (15) samples by NaI (Tl) at the Nuclear Materials Authority, Egypt. The NaI detector advantages are its low cost, linear energy response, and flexibility in size and shape, while the main disadvantages are its poor energy resolution and response time. The best detector resolution is about 6% when measuring the 662 keV gamma ray from 137Cs. It is sealed assembly, which includes a NaI (Tl) crystal, coupled to PC-MCA Canberra Accuses. The cylindrical Pb shield detector (100 mm thick) with shielded fixed bottom and movable cover was used to reduce gamma ray background. This Pb shield contains an inner concentric cylinder of Cu (0.3 mm thick) in order to absorb the generated X-rays in the lead shield. The background distribution in the environment around the detector was determined using the same empty specification package, which was counted in the same manner and in the same geometry as the samples.

## 4. Petrography

Representative samples of the Homrit Waggat granitic rocks are investigated petrographically using a polarizing microscope. These rocks can be classified based on the mineralogical constituents into granodiorite, tonalite, syenogranite, alkali-feldspar granite, and albitized granite. Petrographic descriptions for these rock types are given below from the oldest to the youngest.

**Tonalite** is coarse-grained and comprised mainly of plagioclase, quartz, and biotite with a minor amount of potash feldspar. Two generations of quartz occur. The former is coarse-grained; they are fractured and exhibit undulose extension. The second is fine-grained and mostly fills the fracture between other constituents. Plagioclase occurs as tabular and coarse-grained and reveals extensive turbidity (saussuritization) surface as a result of deformation processes. Fine-grained hornblende occurs and is completely altered to chlorite.

**Granodiorite** is represented by medium grain with hypidiomorphic texture. It comprises plagioclase, K-feldspar, and quartz as essential minerals, whereas sphene, zircon, and apatite are the main accessories. Plagioclase occurs as tabular, subhedral crystals that reveal lamellar twinning. Occasionally, they show perfect zonation (Figure 2a) with slight turbidity due to kaolinitization and saussritization. K-feldspars are represented by patchy, anhedral, and turbid (kaolinitized) perthite. They are commonly corroded by quartz grains and enclose plagioclase crystals. Quartz occurs as fine-(interstitial)-to-medium grain with anhedral crystals. The main mafic mineral is biotite, which presents as subhedral flakes that are partially to completely altered to chlorite. Well-developed sphene (titanite) occurs in aggregates with sphenoidal (wedge) shape (Figure 2b). Scattered iron oxides also occur and are commonly associated with biotite.

**Syenogranite** is medium-tocoarse-grained with hypidiomorphic texture. These rocks consist mainly of K-feldspar, quartz, plagioclase, and biotite as essential minerals with iron oxide, allanite, and zircon as accessory minerals. K-feldspar is represented by microcline- and orthoclase-perthite. They reveal a slightly dusty surface as a result of kaolinitization and seritization. They are corroded by plagioclase and quartz. Occasionally, K-feldspar engulfs small biotite and plagioclase crystals. Micrographic texture occurs due to intergrowth between K-feldspar and quartz. Quartz exhibits a normal extension that is sometimes enclosed in plagioclase crystals. Plagioclase occurs as tabular and subhedral crystals. Biotite crystals are altered to chlorite and commonly host iron oxides (Figure 2c). Short minutes of zircon crystals are mostly hosted in mafic minerals.

**Alkali feldspar granites** are medium-to-coarse-grained with hypidiomorphic texture. Despite having the same mineralogical constituents of syenogranites, K-feldspars and quartz are the dominant minerals, since they reach up to 85 vol.%. K-feldspars include flamy (Figure 2d) and patchy orthoclase perthite as well as microcline minerals. Microcline shows perfect cross-hatching. Both orthoclase perthite and microcline are kaolinitized and seritized. Poikilitically, they are engulfing plagioclase crystals (Figure 2e). Quartz occurs as fine-to-medium-grained with clear surface. Plagioclase is of albite which is rare and commonly presents as tabular crystals corroded by orthoclase perthite. Biotite is the main mafic mineral that reveals variable alteration along its cleavage planes.

**Albitized granite** is fine-to-medium-grained and composed mainly of quartz and albite as the dominant minerals (up to 60 vol%). K-feldspar occurs with amounts reaching up to 25 vol.%, commonly anhedral crystal filling interstitial space between albite crystals. It is noticeable that all minerals reveal high relief. Albite occurs as subhedral and tabular crystals, sometimes fractured and saussuritized (Figure 2f). These rocks are characterized by abundant interstitial fluorite crystals as accessory minerals. The main mafic mineral is biotite that is slightly altered to chlorite.

## 5. Results and Discussion

### 5.1. Geochemical Characteristics

The geochemistry of Hormit Waggat intrusion was previously examined by many authors [10]. In the present work, we analyzed a few representative samples of Homrit Waggat granite rocks. Bulk rock [(major (%) and trace elements (ppm)] geochemical data and their normative values of multi magmatic Homrit Waggat rocks are given in Table 1.

The examined Homrit Waggat rocks exhibited a wide gap in their composition, certainly in SiO_2_ (65.58% in granodiorite; 76.8% in syenogranites); Al_2_O_3_ (13.34% in syenogranites; 17.02% in granodiorite); K_2_O/Na_2_O ratio, which increased from albitized granites (0.47%) to syenogranites (2.22%); CaO and Fe_2_O_3_^t^ (0.29%, 0.37% in alkali-feldspar granites; 4.17%, 4.71% in granodiorite, respectively). Older granitic rocks (tonalite and granodiorite) have the lowest content of K_2_O/Na_2_O in comparison with younger granitic rocks (syenogranites and alkali-feldspar granites), with the exception of albitized granites due to the enrichment of albite. Tonalite and granodiorite comprise the highest concentrations in Al_2_O_3_, TiO_2_, Fe_2_O_3_^t^, CaO, Sc, Sr, Ba, V, Ni, Cu, and Ab + An but are depleted in TiO_2_, Na_2_O + K_2_O, Rb, and Nb relative to those of syenogranites and alkali-feldspar granites. Among the examined rocks, albitized granites contain the highest concentrations of Na_2_O (4.1%) and Ga (23.6 ppm) as well as Ab + An (41.98%) normative values as a result of albite enrichment.

### 5.2. Classification and Tectonic Setting

Variable classification diagrams are used to discriminate the examined rocks. In terms of total alkalis versus silica (TAS) diagram [18], all of the studied samples lie within the granites field, whereas tonalite and granodiorite fall within the field of granodiorite (Figure 3a). With further constraints, alkali-feldspar and syenogranite samples plot in the same field; albitized granites lie within the monzogranites field; and tonalite-granodiorite falls within the field of granodiorite (Figure 3b), using their calculated normative values [19].

It is noticeable that the studied Homrit Waggat rocks have high-K calc-alkaline affinity, whereas tonalite-granodiorite and albitized granites have medium-K calc-alkaline affinity [20] (Figure 3c). The rocks studied have alumina saturation indices (A/CNK = Al_2_O_3_/CaO + Na_2_O + K_2_O) greater than 1.1 (1.71–2.27), suggesting their peraluminous character [22]. In addition, they possess agpaitic indices (AI) lower than 0.87, indicating calc-alkaline characteristic. Among the Homrit Waggat rocks, tonalite-granodiorite rocks have low K_2_O/MgO content, reflecting I- and S-type affinity (Figure 3d), while other rock samples occupy A-type field [21].

It is noticeable from multi trace elements that are normalized to primitive mantle [23] that tonalite-granodiorite patterns reveal enrichment of low-field-strength elements (LFSEs) relative to high-field-strength-elements (HFSEs), with positive Ba, Pb, and Sr anomalies as well as negative Nb anomaly, reflecting subduction affinity (Figure 4a). On the contrary, other granitic types exhibit strong positive Rb anomaly without negative Nb anomaly. In addition, they reveal strong Ba, Sr, and Ti negative anomalies (Figure 4b), suggesting fractional crystallization of feldspars and titanite minerals.

Egyptian granitic rocks have different tectonic setting due to the difference in their mineralogical as well as geochemical composition. The studied tonalite-granodiorite rocks have geochemical characteristics of subduction-related regime. Therefore, they locate in volcanic arc granites (Figure 4c) in the binary diagram [24]. Controversially, Homrit Waggat younger granites (alkali-feldspar, syenogranites, and albitized granites) fall in the post-collisional and within plate granites. The same results have been obtained by using a ternary diagram [10,26], where the examined rocks lie rather than in the field of older calc-alkaline (I, metagabbro and diorite). The examined tonalite-granodiorite locate in the early phase of younger granites (II, granodiorite-tonalite-tonalite associations), whereas alkali-feldspar and syenogranites lie in the III (Figure 4d) field of late- to post-collisional setting (later phase of calc-alkaline granites). Albitized granites locate near the late- to post-collisional setting, and their origin may be related to albitic metasomatism of different granitic rocks [10].

### 5.3. Radiometric and Environmental Hazard Indices

The mean values of ^238^U, ^232^Th, ^226^Ra, and ^240^K activity concentrations (Bq/kg) are given in Table 2. It is noticeable that the mean values of ^238^U, ^232^Th, ^226^Ra, and ^240^K for alkali-feldspar (49.6, 44.44, 44.4, and 936 Bq/kg, respectively) are higher than the world safety limits of [27]. In addition, ^238^U concentration in albitized granites, granodiorite, and tonalite is under detection limits.

The U, Th, and K concentrations of El-Missikat area have averages of 257.2 Bq kg^−1^, 111 Bq kg^−1^, and 832.2 Bq kg^−1^ for monzogranite. For syenogranite, the U, Th, and K concentrations average 176.5, 92.25, and 899.2 Bq kg^−1^, respectively [15].

On the other hand, the granitic rocks of Um Taghir area have an average activity (range) of ^238^U series, ^232^Th series, and ^40^K of 15.7 (8.7–25.7), 13.2 (4.7–11.7), and 703.8 (195.1–1371.8) Bq/kg, respectively. The average concentration values of ^238^U and ^232^Th are lower than those of the world’s average and the average activities in the present study [28].

Some radiological hazards were applied for the examined samples in order to deduce their impact on human organs. They include annual effective dose (AED), gamma radiation index (Iγ), internal (H_in_) and external (H_ex_) indices, absorbed gamma dose rate (D), and radium equivalent activity (Ra_eq_) (Table 2).

**Radium equivalent activity** (Ra_eq_) refers to the external dose from exposure to gamma rays and the internal dose from exposure to alpha particles. This index is directly related to the external and internal gamma dose from radon and its progenies [29]. Ra_eq_ is a proper index that uses activity concentrations of ^226^Ra, ^232^Th, and ^40^K in the following equation [2,30,31];
Ra_eq_ (Bq kg^−1^) = ^226^Ra + 1.43 ^232^Th + 0.077 ^40^K(1)

The calculated average values of the radium equivalent index for the studied samples collected from Homrit Waggat area were recorded as 180, 154, 107.67, 90.13, and 42.66 Bq kg^−1^ for alkali-feldspar granites, syenogranites, albitized granites, granodiorite, and tonalite, respectively. The obtained results for all studied samples observed are less than the maximum value of 370 Bq kg^−1^ for radium equivalent index recommended value [15].

Radium equivalent content in El Sela area has an average of 734.3 Bq/Kg, which was significantly higher than the maximum permitted value in the world’s average and the average activities in the present study [32]. The average range of radium equivalent of the granitic rocks was 319.8 Bq kg^−1^, and the average (range) of radium equivalent for red and black jasperoid veins was found to be 1823.6 Bq kg^−1^. The estimated average value of red and black jasperoid veins of radium equivalent in El-Missikat area is higher than that of the recommended maximum value [15].

### 5.4. Excess Lifetime Cancer Risk (ELCR)

Excess lifetime cancer risk (ELCR) was calculated using the following equation and presented in Table 2.
ELCR = AEDE * DL (70y) * Rf (0.5 SV^−1^)(2)
where AEDE, DL, and Rf are the annual effective dose equivalent, duration of life (70 y), and risk factor (SV^−1^), fatal cancer risk per Sievert. For stochastic effects, ICRP 60 uses values of 0.05 for the public [33].

The calculated value of ELCR ranges from 0.00001 to 0.00004 in the investigated rocks. This value of ELCR was less than the world average [27].

The granitic rocks of El-Missikat area have excess lifetime cancer risk with average reach of about 633.0, while red and black jasperoid veins have an average of up to 3592, such that the granitic rocks of El-Missikat area are exceeding the world permissible safe criteria and are considered a risk source for human environment [15].

Likewise, **absorbed gamma dose rate** is defined as the energy transported by ionizing radiation to the mass unit of matter. It can be estimated from the radionuclide distribution in the air at 1 m above sea level as a result of gamma radiations [2,27,34,35] using ^226^Ra, ^232^Th, and ^40^K activity concentrations as in below:D (nGy h^−1^) = 0.462 ^226^Ra + 0.604 ^232^Th + 0.0417 ^40^K(3)

The calculated average values of absorbed gamma dose rate are 86.38, 75.93, 53.49, 43.6, and 20.84 nGy h^−1^ for the examined alkali-feldspar granites, syenogranites, albitized granites, granodiorite, and tonalite, respectively. It was noticed that the alkali-feldspar granite and syenogranite absorbed dose is higher than the recommended value (57 nGy h^−1^) [34], whereas albitized granites, granodiorite, and tonalite are low.

In addition, the **internal radiation index** (H_in_) (Figure 5a) can be determined in order to deduce the radon impact on respiratory organs. The H_in_ can be calculated using the following equation:H_in_ = ^226^Ra/185 + ^232^Th/259 + ^40^K/4810 < 1(4)

Values calculated of the H_in_ vary from 0.15 (tonalite) to 0.61 (alkali-feldspar granites), which are lower than the acceptable level [2,27,34,35].

Moreover, **external hazard index** (H_ex_) (Figure 5b) is used in order to assess the radiation dose as below [2,27,34,35].
H_ex_ = ^226^Ra/370 + ^232^Th/259 + ^40^K/4810 ≤ 1(5)

In terms of H_ex,_ the calculated values vary from 0.12 (tonalite) to 0.49 (alkali-feldspar granites), which are lower than the acceptable level [16,36].

In addition, **γ-radiation hazard (Iγ)** is used to evaluate the radiation hazard accompanied with natural radionuclides in the examined samples. Gamma radiation is defined as in the following equation [2,27]:Iγ = ^226^Ra/300 + ^232^Th/200 + ^40^K/4000(6)

It is noticed that the Iγ ranges from 0.14 (tonalite) to 0.6 (alkali-feldspar granites), which falls within the acceptable levels [16,27].

**Annual effective dose** (AED) can be estimated using the absorbed dose values with the conversion factor of 0.7 Sv/Gy and the outdoor occupancy factor of 0.2 [2,16].
AED (mSv/y) = D(nGy/h) × 8760 (h) × 0.2 × 0.7 (Sv/Gy) × 10^−6^(7)

AED obtained values ranging from 0.03 (tonalite) to 0.11 (alkali-feldspar granites), which falls within the acceptable levels [2,16,36].

From the previous parameters, the obtained values of the examined rocks indicate that these rocks are safe to be used as ornamental stones or in other applications such as the ceramic industry.

## 6. Conclusions

Homrit Waggat granitic rocks represent a post-collisional intrusion of ring-like shape which consists of two main phases of magmatic activity. The early phase includes tonalite-granodiorite suite of calc-alkaline nature and subduction-related setting. Controversially, the latter phase includes alkali-feldspar granite, syenogranite, and albitized granite with high-K calc alkaline and related to post-orogenic granites. They exhibit strong positive Rb anomaly without negative Nb anomaly and strong Ba, Sr, and Ti negative anomalies. Variable hazard indices such as radium equivalent activity (Raeq), annual absorbed gamma dose rate (D), effective dose (AED), gamma radiation index (Iγ), and internal (Hin) and external (Hex) indices were applied for Homrit Waggat rock units. The result of these indices falls within the acceptable standard values of worldwide limits. Thus, harmful radiation effects as a result of the natural radioactivity of investigated rocks do not pose a danger to the public in the field of construction and adornment or in the extent of their impact on those who use them in their homes or for other purposes.

## Figures and Tables

**Figure 1 materials-15-04069-f001:**
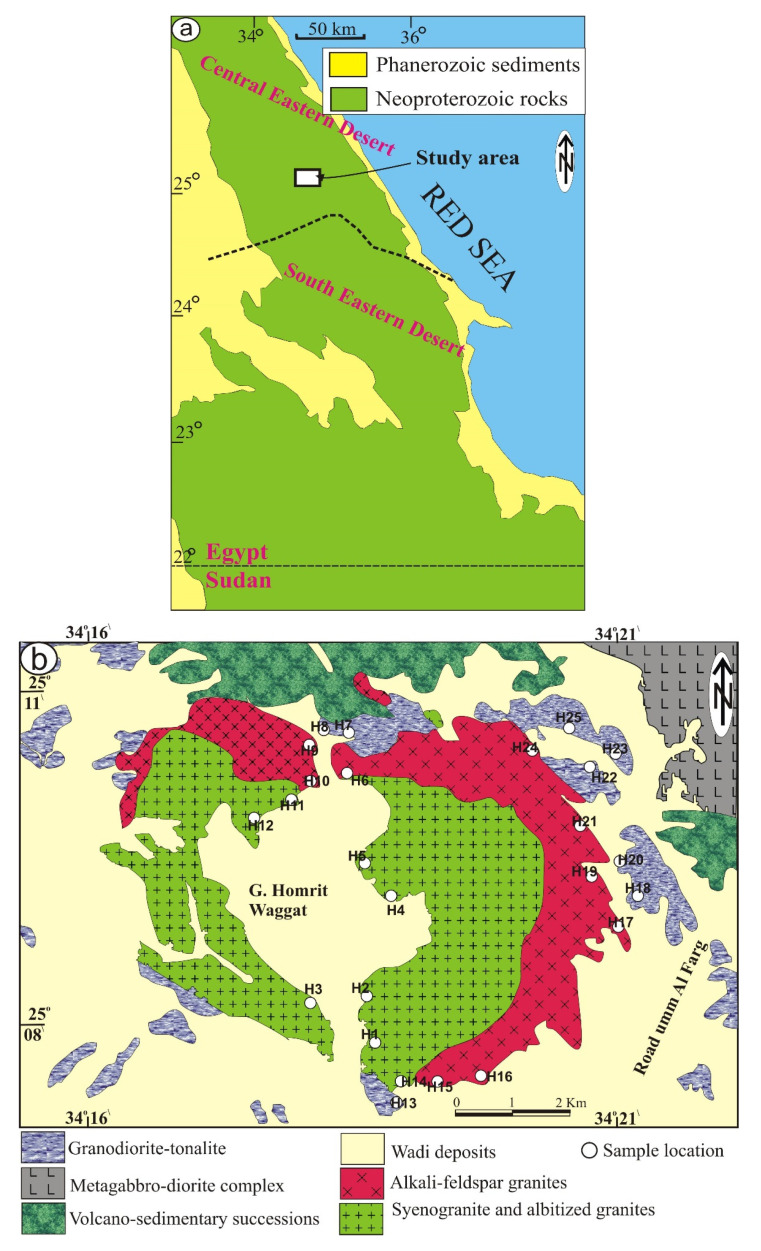
(**a**) Location map of Homrit Waggat area, Central Eastern Desert of Egypt; (**b**) geologic map of Homrit Waggat area modified after [10].

**Figure 2 materials-15-04069-f002:**
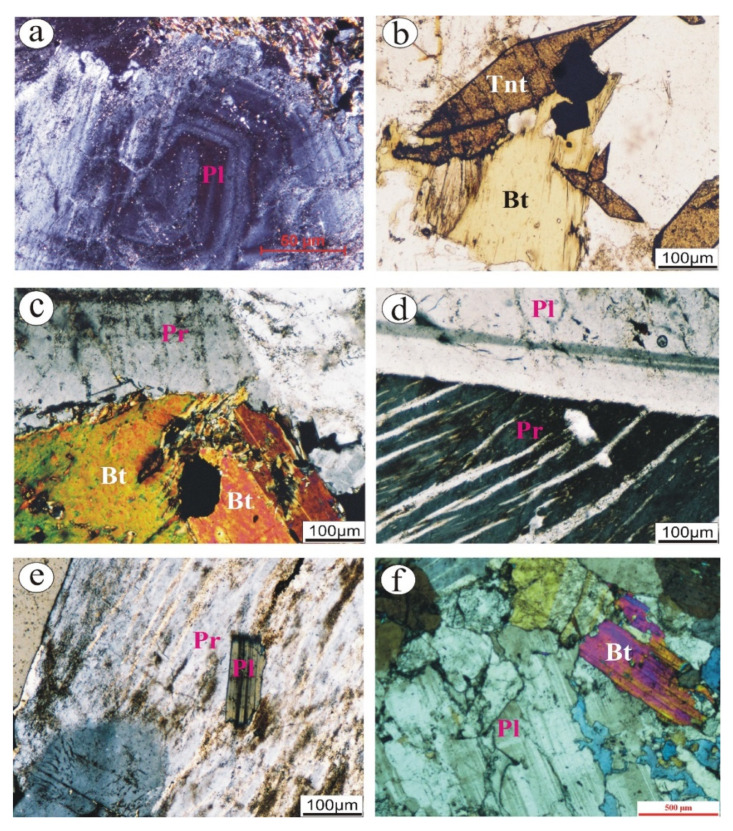
Photomicrographs showing: (**a**) well-developed zonation of plagioclase (Pl) in granodiorite; (**b**) aggregates of wedge-like shape of titanite (Tnt) crystals in granodiorite; (**c**) biotite (bt) is partially altered to chlorite in syenogranites; (**d**) flamy perthite (Pr) associated with plagioclase (Pl) crystal in alkali-feldspar granites; (**e**) turbid perthite engulfing plagioclase crystal in alkali-feldspar granites; and (**f**) fractured albitic plagioclase associated with biotite in albitized granites.

**Figure 3 materials-15-04069-f003:**
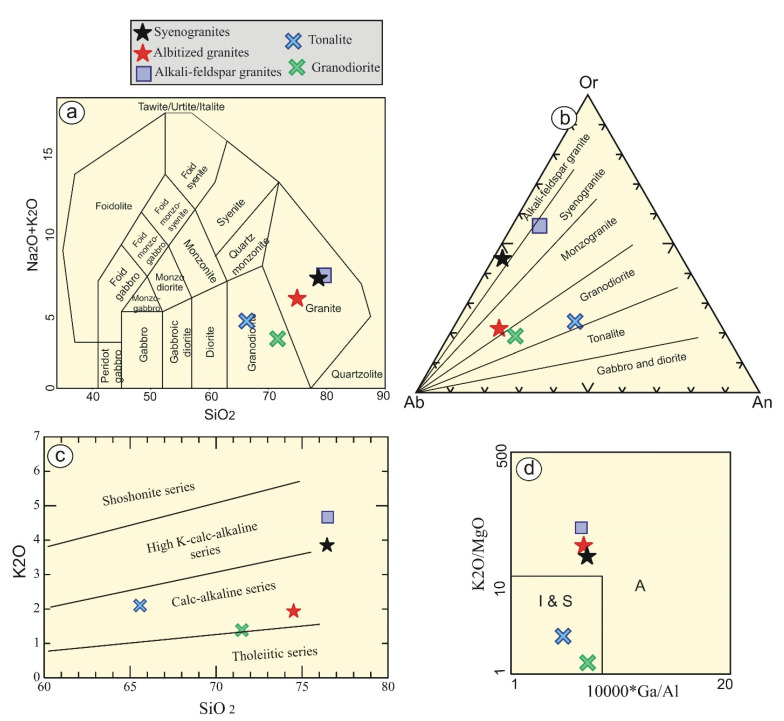
Classification and magma-type diagrams of Homrit Waggat rocks using: (**a**) SiO_2_ vs. Na_2_O + K_2_O diagram [18]; (**b**) Ab-Or-An normative diagram [19]; (**c**) SiO_2_ vs. K_2_O diagram [20]; and (**d**) 10,000*Ga/Al vs. K_2_O/MgO diagram [21].

**Figure 4 materials-15-04069-f004:**
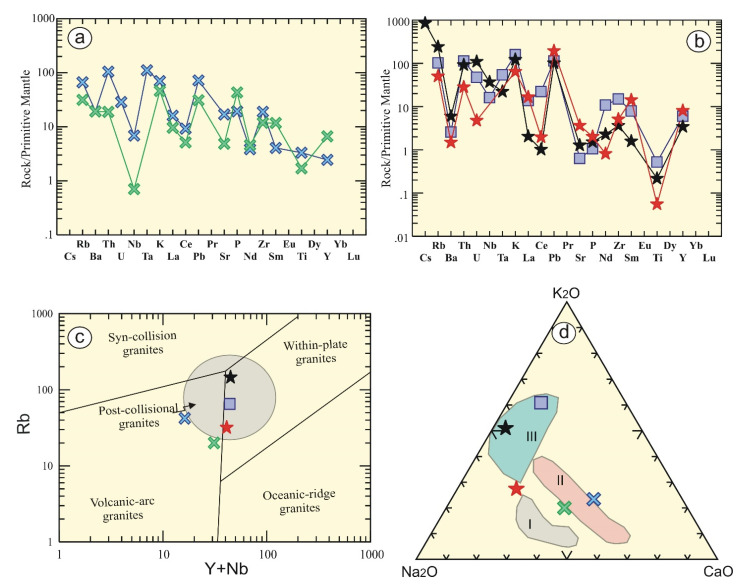
Multi trace elements pattern normalized to primitive mantle [23] for (**a**) tonalite-granodiorite and (**b**) alkali-feldspar granites, syenogranites, and albitized granites. Tectonic discrimination diagrams (**c**) Y + Nb versus Rb binary diagram [24], where post-collisional field is adapted from [25]; and (**d**) NKC diagram (Na_2_O-K_2_O-CaO) of the Egyptian granitoids [26], III = late (subphase) orogenic calc-alkaline phase, II = early subphase of younger granites, I = calc-alkaline diorite and metagabbro. Rock symbols as in Figure 3.

**Figure 5 materials-15-04069-f005:**
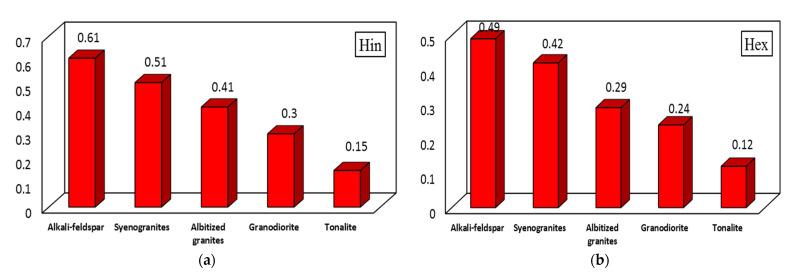
Hazard indices values. (**a**) Internal hazard index value (H_in_) and (**b**) external hazard index value (H_ex_).

**Table 1 materials-15-04069-t001:** Whole-rock analyses (major (%) and trace elements (ppm)) of the examined rocks and their normative values.

Rock Units	Syenogranites	Alkali- Feldspar Granites	Albitized Granites	Granodiorite	Tonalite
SiO_2_	76.803	76.565	74.511	65.585	71.507
Al_2_O_3_	13.349	14.923	16.137	17.024	15.471
TiO_2_	0.113	0.047	0.012	0.717	0.369
Fe_2_O_3_	1.746	0.373	0.548	4.719	3.448
MgO	0.027	0.106	0.028	1.526	0.562
Na_2_O	2.15	3.17	4.109	2.575	2.766
K_2_O	4.781	3.679	1.934	2.102	1.385
CaO	0.844	0.291	1.511	4.174	2.673
MnO	0.042	0.066	0.077	0.059	0.123
P_2_O_5_	0.023	0.033	0.044	0.42	0.94
LOI	0.64	0.62	0.48	0.45	1.42
Total	100.52	99.87	99.39	99.35	100.66
SC	ND	ND	2.4	7.3	5.7
V	ND	ND	ND	27.4	15.1
Co	ND	ND	ND	ND	ND
Ni	ND	ND	ND	6.2	ND
Cu	3.5	3.20	3	6.4	4.5
Zn	48.1	51.10	9.7	56.5	43.9
Ga	20.2	23.30	23.6	19.9	14.8
Rb	65.20	153.300	31.9	42.2	20
Sr	13.40	27.00	76.9	356.2	102.4
Zr	168.50	40.70	57.1	212	132.6
Y	27.30	15.50	36.7	11.1	30.2
Nb	11.5	26.40	ND	4.9	0.5
Cd	ND	2.00	ND	ND	3.10
Sn	8.7	10.9	6.8	8.4	10.5
Cs	ND	6.7	ND	ND	ND
Ba	18	41.7	10.4	135.2	133.9
La	9.5	ND	11.4	11	6.6
Ce	39.7	ND	3.5	16.4	9.1
Nd	14.7	3.1	ND	5.2	6.2
Sm	3.5	ND	6.3	ND	5.2
Hf	5.9	5.8	3.6	5.1	7.3
Ta	2.2	ND	ND	4.5	ND
W	2.4	0.7	2.1	15.5	13
Tl	7.8	6.4	8.6	7.8	7.8
Pb	8.1	7.2	13.8	5.1	2.2
Th	9.6	7.8	2.4	8.9	ND
U	ND	2.3	ND	ND	ND
Qz	44.2	43.4	40.06	32.54	49
Crn	3.2	5.28	5	4	9
Or	28.3	21.74	11	12	8
Ab	18.2	26.82	35	22	23.41
An	4.0	1.23	7	18	12
Hy	0.1	0.3	0.07	3.80	1
Hm	2	0	0	5	3

ND indicates that the measuring system does not give any concentrations of these elements.

**Table 2 materials-15-04069-t002:** The measured activity concentrations for radionuclides and radiological hazard indices for the examined rocks.

Activity Concentration (Bq/Kg)						Radiological Hazard Indices						
Rock Units		238 U	232 Th	226 Ra	240 K	Raeq	Absorbed	Hin	Hex	Iγ	Eff. Dose	ELCR * 10^−4^
**Alkali-feldspar granite**		41.60	36.44	36.40	927.87	159.96	77.52	0.53	0.43	0.54	0.10	0.00004
		51.60	46.44	46.40	937.87	185.03	88.60	0.63	0.50	0.62	0.11	0.00003
		55.60	50.44	50.40	941.87	195.05	93.03	0.66	0.53	0.66	0.11	0.00004
	**Av**	49.60	44.44	44.40	935.87	180.01	86.38	0.61	0.49	0.60	0.11	0.00004
**Syenogranites**		54.00	20.28	25.30	1034.29	133.94	67.07	0.43	0.36	0.44	0.08	0.00004
		64.00	30.28	35.30	1044.29	159.01	78.14	0.52	0.43	0.53	0.10	0.00003
		68.00	34.28	39.30	1048.29	169.04	82.58	0.56	0.46	0.56	0.10	0.00003
	**Av**	62.00	28.28	33.30	1042.29	154.00	75.93	0.51	0.42	0.51	0.09	0.00003
**Albitized granites**		0.00	2.03	36.40	724.42	95.08	48.25	0.36	0.26	0.31	0.06	0.00004
		0.00	5.02	46.40	734.42	110.13	55.09	0.42	0.30	0.36	0.07	0.00002
		0.00	5.07	50.40	738.42	114.51	57.14	0.45	0.31	0.38	0.07	0.00002
	**Av**	0.00	4.04	44.40	732.42	106.57	53.49	0.41	0.29	0.35	0.07	0.00002
**Granodiorite**		0.00	12.20	14.20	499.06	70.07	34.74	0.23	0.19	0.23	0.04	0.00002
		0.00	22.20	24.20	509.06	95.14	45.82	0.32	0.26	0.32	0.06	0.00002
		0.00	26.20	28.20	513.06	105.17	50.25	0.36	0.28	0.35	0.06	0.00001
	**Av**	0.00	20.20	22.20	507.06	90.13	43.60	0.30	0.24	0.30	0.05	0.00002
**Tonalite**		0.00	0.08	3.10	251.79	22.60	11.98	0.07	0.06	0.07	0.01	0.00002
		0.00	10.08	13.10	261.79	47.67	23.06	0.16	0.13	0.16	0.03	0.00001
		0.00	14.08	17.10	265.79	57.70	27.49	0.20	0.16	0.19	0.03	0.00001
	**Av**	0.00	8.08	11.10	259.79	42.66	20.84	0.15	0.12	0.14	0.03	0.00001

## Data Availability

The data presented in this study are available on request from the corresponding author.

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
