# Peer review of "Radiological Hazards and Natural Radionuclide Distribution in Granitic Rocks of Homrit Waggat Area, Central Eastern Desert, Egypt"

_materials, 2022, doi:10.3390/ma15124069_

Round 1
Reviewer 1 Report
The materials analyzed are well described as well as the methods. For Figure 4 should appear a legend, because the reader should clearly see the meaning of it. Some references addition, also for equations it will be very useful.
The paper is well written. A agree with the publication with some minor changes.

Author Response
Respond to Reviewer comments
Title: Radiological hazards and natural radionuclide distribution in granitic rocks of Homrit Waggat area, Central Eastern Desert, Egypt
Manuscript Number: 1688606
The authors would like to thank Editor and Reviewer 1 for their comments and recommendations that have helped to improve the clarity, readability and academic standing of this paper. Below are detailed responses with extracts showing the way of addressing reviewer comments.
Reviewer 1
The materials analyzed are well described as well as the methods. For Figure 4 should appear a legend, because the reader should clearly see the meaning of it. Some references addition, also for equations it will be very useful. The paper is well written. A agree with the publication with some minor changes.
Response: All comments in the attached file have been done.

Reviewer 2 Report
This manuscript is about granite rocks' radiological analysis. I have several fundamental problems with it.
- although the aim is clear, I do not see clearly how it has been achieved in the work
- material and methods: not enough detailed, the method description and for example, the number of samples, the methods' parameters are missing.
- results: confusing and difficult to understand in several places. (for example, Table 1.)
- description of the radiological hazard indices is in the results, but it is wrong, its place is in the Material and methods
- the conclusion is very poor.
Author Response
Respond to Reviewer comments
Title: Radiological hazards and natural radionuclide distribution in granitic rocks of Homrit Waggat area, Central Eastern Desert, Egypt
Manuscript Number: 1688606
The authors would like to thank Editor and Reviewer 1 for their comments and recommendations that have helped to improve the clarity, readability and academic standing of this paper. Below are detailed responses with extracts showing the way of addressing reviewer comments.
Reviewer 2
This manuscript is about granite rocks' radiological analysis. I have several fundamental problems with it.
- Although the aim is clear, I do not see clearly how it has been achieved in the work.
Response: Detection the natural radioactivity as well as radiological health risk of different Homrit Waggat magmatic rocks, reflecting that these rocks can be used as ornamental stone or decorative materials. This is discussed in section 5.2.
- Material and methods: not enough detailed, the method description and for example, the number of samples, the methods' parameters are missing.
Response: Analysis description and number of samples have been added in section 3.
- Results: confusing and difficult to understand in several places. (for example, Table 1.)
Response: This table contain bulk (whole) rock [(major (%) and trace elements (ppm)] geochemical data and their normative values (Streckeisen, 1996) of multi magmatic Homrit Waggat rocks. The geochemical analysis (few samples) just used to support the petrographic examination.
- Description of the radiological hazard indices is in the results, but it is wrong, its place is in the Material and methods
Thanks for this comment
Response: Therefore, the paper was formatted in this way so as not to confuse the reader later. It has been updated
- The conclusion is very poor.
Response: Done

Reviewer 3 Report
Dear Editor
I have checked this manuscript. The topic is interesting. However, unfortunately, the authors are more focused on petrography, petrology, and tectonic setting than on the topic of the manuscript. Only minor chemistry discussion is enough. Delete most parts of the petrography, petrology, and tectonic setting. These parts are out of your subject. In addition, the number of data is too less to talk about petrology and geodynamics. Consequently, more focus on radiation data needs it. Here just I found a few simple calculations based on some clear equations. More focus on the isotope radiation abundances and half time should be done. In addition, some environmental measurement is needed to confirm the author's idea.
Please added values of effective dose (AED), gamma radiation index (Iγ), internal (Hin) and external (Hex) indices, absorbed gamma dose rate (D), and Radium equivalent activity (Req) for your samples and also the acceptable worldwide limits values in the abstract.
Add some granitic bodies which have higher radiation and are known as the unsafe area in the introduction. And also make some correlation between the unsafe and safe granites in the text.
The location of the sampling should be shown in figure 2.
The number of the samples is too less and it cannot cover such big bodies.
Best wishes
Author Response
Respond to Reviewer comments
Title: Radiological hazards and natural radionuclide distribution in granitic rocks of Homrit Waggat area, Central Eastern Desert, Egypt
Manuscript Number: 1688606
The authors would like to thank Editor and Reviewer 1 for their comments and recommendations that have helped to improve the clarity, readability and academic standing of this paper. Below are detailed responses with extracts showing the way of addressing reviewer comments.
Reviewer 3
I have checked this manuscript. The topic is interesting. However, unfortunately, the authors are more focused on petrography, petrology, and tectonic setting than on the topic of the manuscript. Only minor chemistry discussion is enough. Delete most parts of the petrography, petrology, and tectonic setting. These parts are out of your subject. In addition, the number of data is too less to talk about petrology and geodynamics.
Response: We discuss in detail the petrographic (mineralogical composition) description of the examined rocks in order to classify them according to their time of emplacement (geochronological sequence, older or younger). In addition, the geochemistry section including geochemical characteristics, classification, and tectonic setting of the examined rocks have been manifested by Azer et al., 2020. However, here, we just selected some representative sample to link the mineralogical (petrography) with geochemical composition of these rocks. The new is natural radioactivity detection in these rocks. Therefore, most indices have been applied for the examined rocks.
Consequently, more focus on radiation data needs it. Here just I found a few simple calculations based on some clear equations. More focus on the isotope radiation abundances and half time should be done. In addition, some environmental measurement is needed to confirm the author's idea.
Response: It have been done
Please added values of effective dose (AED), gamma radiation index (Iγ), internal (Hin) and external (Hex) indices, absorbed gamma dose rate (D), and Radium equivalent activity (Req) for your samples and also the acceptable worldwide limits values in the abstract.
Response: have been done
Add some granitic bodies which have higher radiation and are known as the unsafe area in the introduction. And also make some correlation between the unsafe and safe granites in the text.
Response: Done
The location of the sampling should be shown in figure 2.
Response: Done
The number of the samples is too less, and it cannot cover such big bodies.
Response: Number of samples have been manifested in section 3. Twenty five samples were collected from field, and fifteen samples were selected for petrographic investigations (to identify the variable mineralogical composition) using polarizing microscope. Just five representative samples have been selected for bulk-rock geochemical analysis to support the petrographic examination. In addition, natural radioactivity for fifteen samples were performed using NaI technique that listed in supplementary material (we used the mean value in table 2).

Round 2
Reviewer 1 Report
All comments have been completed. The study is well documented and the results very well presented. I recommend the publication of the article.
Author Response
The authors would like to thank Editor and reviewers for their comments and recommendations that have helped to improve the clarity, readability and academic standing of this paper. Below are detailed responses with extracts showing the way of addressing reviewer's comments.

Reviewer 2 Report
I do not have more comments
Author Response

(The authors gave the same response as above.)

Reviewer 3 Report
1. In figure 1 please add the sample name
2. some digits are not true In table 1
TRANSLATE with x English
| Arabic | Hebrew | Polish |
| Bulgarian | Hindi | Portuguese |
| Catalan | Hmong Daw | Romanian |
| Chinese Simplified | Hungarian | Russian |
| Chinese Traditional | Indonesian | Slovak |
| Czech | Italian | Slovenian |
| Danish | Japanese | Spanish |
| Dutch | Klingon | Swedish |
| English | Korean | Thai |
| Estonian | Latvian | Turkish |
| Finnish | Lithuanian | Ukrainian |
| French | Malay | Urdu |
| German | Maltese | Vietnamese |
| Greek | Norwegian | Welsh |
| Haitian Creole | Persian |
TRANSLATE with EMBED THE SNIPPET BELOW IN YOUR SITE Enable collaborative features and customize widget: Bing Webmaster Portal Back
Author Response

(The authors gave the same response as above.)
